# Primary Cutaneous Cryptococcosis in an Immunocompetent Patient: Diagnostic Workflow and Choice of Treatment

**DOI:** 10.3390/diagnostics13193149

**Published:** 2023-10-07

**Authors:** Francesca Panza, Francesca Montagnani, Gennaro Baldino, Cosimo Custoza, Mario Tumbarello, Massimiliano Fabbiani

**Affiliations:** 1Department of Medical Biotechnologies, University of Siena, 53100 Siena, Italy; francesca.panza@unifi.it (F.P.); francesca.montagnani@unisi.it (F.M.); mario.tumbarello@unisi.it (M.T.); 2Infectious and Tropical Diseases Unit, Azienda Ospedaliero—Universitaria Senese, 53100 Siena, Italy; 3Department of Health Promotion Sciences, Maternal and Infant Care, Internal Medicine and Medical Specialties (PROMISE), University of Palermo, 90133 Palermo, Italy; gennarobld@hotmail.it; 4Pathological Anatomy Unit, Department of Medical, Surgical and Neurological Science, University of Siena, 53100 Siena, Italy; cosimo.custoza@outlook.it

**Keywords:** *Cryptococcus neoformans*, primary cutaneous cryptococcosis, immunocompetent host, skin ulcer, fluconazole, fungal infection

## Abstract

Cryptococcosis is an opportunistic infection in immunocompromised patients, involving mainly the lungs and central nervous system; however, the skin, eyes and genitourinary tract could also be involved as secondary sites of infection. Primary cutaneous cryptococcosis (PCC) is a distinct clinical entity that can occur in both immunocompetent and -compromised patients, usually trough skin injury. In immunocompetent patients, it is a very rare infection, presenting with non-specific clinical pictures and being challenging to diagnose. Herein, we present the case of an immunocompetent man with PCC due to *Cryptococcus neoformans* on his right forearm. PCC was diagnosed by a histological and cultural examination. Causes of concomitant immunosuppression were ruled out. A secondary cutaneous cryptococcosis was excluded with careful investigations. Therapy with oral fluconazole for three months was successfully performed, without evidence of recurrence in the following six months. Complete clinical recovery was achieved after three months of oral antifungal therapy, suggesting that longer courses of treatment could be avoided when faced with PCC in immunocompetent patients.

A 72-year-old man, with no history of other diseases except for psoriasis and colic diverticulosis, presented to our department with a painful erythematous area with rounded ulcers on his right forearm lasting for 5 months. Before lesions development, he reported an accidental unspecified minor skin trauma while working in the countryside. The patient was initially evaluated by a dermatologist who carried out several lesion swabs for bacterial cultures, leading to the isolation of a methicillin-resistant Staphylococcus aureus (MRSA) in two out of multiple samples. Suspecting a bacterial soft tissue infection, the patient was treated with different consecutive antibiotic regimens (trimethoprim/sulfamethoxazole + rifampicin, minocycline, intramuscular teicoplanin for 4 weeks and then oral linezolid for 6 days) without any benefit. At the time of our evaluation, the lesion was worsening since the first dermatologic visit: a major irregular ulcer (5 × 3 cm in diameter) with elevated borders and a central loss of substance was accompanied by approximately six additional minor satellite ulcers. The skin around the ulcers was diffusely erythematous and edematous, and no signs of necrosis or bleeding could be appreciated (Figure 1a). There was no regional lymphadenopathy and no other skin involvement or systemic symptoms. The patient was not taking any immunosuppressive drug and had no history of any evident contact with bird droppings, pigeons, or other animals, and denied recent traveling. At the blood tests, C reactive protein level was 0.95 mg/dL (normal values 0.00–0.50 mg/dL), and the total lymphocyte count, CD4 and CD8 lymphocyte count, and blood immunoglobulin levels (total and different subclasses) were normal. HIV serology was negative. Hematological malignancies were also ruled out. We performed lesion swab cultures for bacteria and fungi and, after 48 h of incubation, a lot of mucoid colonies were noted on Sabouraud Dextrose Agar + Chloramphenicol (CAF) and were identified via standard microbiological procedures as *Cryptococcus neoformans*. An incisional biopsy of the lesion for histological and cultural examination was also performed (see Figure 2). The cultural examination of the biopsy confirmed the presence of *C. neoformans.* Cultures and direct DNA amplification via polymerase chain reaction for mycobacteria resulted negative. To exclude a disseminated cryptococcal infection, further investigations were carried out by performing blood cultures, serum cryptococcal antigen detection and chest X-ray and abdomen ultrasound, all of which resulted negative. Since the patient did not show any neurological symptoms and the serum cryptococcal antigen was negative, a lumbar puncture was not performed. Because of these results, a diagnosis of PCC on an immunocompetent patient was made. Treatment with oral fluconazole 400 mg once daily was started. The patient remained always apyretic and in good general conditions. During the monthly follow-up, the skin lesion gradually improved, and routine blood examinations did not show any alteration. After three months, the skin ulcers and the signs of skin inflammation completely resolved; therefore, the antifungal therapy was stopped. In the following 6 months, no evidence of disease recurrence was revealed (Figure 1b).

PCC is a clinical entity that occurs after transcutaneous entry of the yeast through a skin lesion [1]. In the literature, PCC is defined by the identification of *C*. *neoformans* in a biopsy skin specimen or by culture, together with clinical criteria and in absence of dissemination [2]. A documented history of skin injury or outdoor activities predisposing to wounds have generally been associated with the risk of PCC disease [3]. In this report, the patient reported an unspecific traumatic minor injury while working in the countryside some months before the disease’s onset.

Usually, the PCC lesion is a solitary lesion that appears on unclothed parts of the body [4]. The most frequently involved sites are the extremities, especially the upper ones. Forearm, hands and fingers are particularly common sites of infection in immunocompetent patients, while the trunk or the lower extremities are more common in AIDS patients [5]. Skin lesions could present in many different clinical pictures: vesicles, nodules, ulcers, pustules, acneiform lesions, granulomas, infiltrating plaques or cellulitis [6]. Cellulitis, ulceration and abscess are the most common clinical features during PCC [7] in immunocompetent patients. In our patient, the skin lesion was clinically in agreement with the literature description of PCC: it was a major lesion accompanied by additional smaller satellite ulcers, located on an uncovered part of the body, and had an ulcerated pattern.

Although cutaneous manifestations of PCC usually differ from secondary cryptococcosis, it must be remembered that almost any type of lesion can be seen during disseminated cryptococcosis. Therefore, the exclusion of a secondary cryptococcosis with careful investigations is always mandatory. Moreover, it should also be considered that primary skin lesions could be the source for the occurrence of systemic cryptococcosis. For this reason, further investigations were performed in our patient after the identification of *C. neoformans*: serum cryptococcal antigen detection, chest radiography, abdomen US and blood cultures to rule out a systemic infection. Lumbar puncture was not performed in this case since there were no signs or symptoms of central nervous system involvement, the serum antigen detection was negative and there was no other evidence of central nervous system nor systemic dissemination. Moreover, patients with *C. neoformans* skin lesions must be thoroughly investigated for possible immunodeficiency conditions; a cellular immune defect, such as HIV infection, or other severe forms of lymphocytopenia could be discovered in this setting [8]. In our report, HIV antibodies and HIV p24 antigen were negative, a complete humoral and cellular immunity evaluation did not show any deficit and led us to diagnose PCC on an immunocompetent patient.

In our patient, there was a diagnostic delay because fungal tests were not initially performed. Medical mycology diagnostic procedures are widely available and not difficult to perform. Anyway, a fungal infection is rarely suspected, and it is often misdiagnosed in immunocompetent hosts. Indeed, it should be highlighted that a fungal infection should also be considered when facing protracted skin lesions not improving after antibacterial treatment.

A standard treatment for PCC in immunocompetent hosts has not been fully defined by guidelines, due to the relative rarity of cases. Fluconazole, itraconazole and amphotericin B are the most commonly used drugs to treat PCC [9]. Amphotericin B (0.5–1 mg/kg/day), combined or not with flucytosin, was the standard therapy for any clinical form of cryptococcosis before the appearance of the azoles. Nowadays, the azole compounds (fluconazole, itraconazole) have become the primary regimen for PCC due to their efficacy against *C. neoformans* and to their availability as an oral formulation. Fluconazole at a dosage of 6 mg/kg/die (usually 400 mg QD) is the most prescribed treatment in patients with single skin lesions, and it is effective in most cases [10]. Itraconazole (dose range 100–400 mg/day) is also a common therapeutic option, but the use of this drug could be limited by a lower tolerability and a wide inter-individual pharmacokinetic variability, also related to an increased potential of drug–drug interactions. The duration of antifungal therapy has not been clearly defined. The majority of PCC cases described in literature have been treated with oral fluconazole or itraconazole alone for an average period of 3–6 months [11,12]. In some cases, therapy was given for 3 months but was accompanied by surgical debridement of the lesion [13,14]. In other cases, the drainage alone of the cutaneous lesions was curative [15]. In another case, a higher dose of antifungal oral therapy was administered only for the first month (fluconazole 400 mg), then reduced to half-dose maintenance therapy (fluconazole 200 mg/day) for another 2 months [16]. In one case, oral fluconazole was administered for only 15 days, with an excellent clinical course [14]. Isolated skin lesions healed without any treatment have also been described [17]. Therefore, the choice and duration of therapy seem to be case-dependent. The most determinant factors in the therapeutic strategy for PCC are the localization of skin lesions, their extension and the patient’s general conditions, comorbidities and current concomitant therapies.

In the present case, our patient was treated with oral fluconazole 400 mg/day for a total of 3 months, with clinical benefit and no evidence of recurrence. Surgical debridement was not considered necessary. The duration of the antifungal therapy was limited to 3 months, on the basis of the clinical response and lack of immunosuppressive comorbidities. Ensuring close clinical and laboratory monitoring of the patient by re-evaluating him monthly in an outpatient setting, we did not observe any recurrence within the subsequent 6 months.

In conclusion, PCC could be a difficult clinical entity to diagnose. Herein, we gathered evidence that it is important to perform in-depth diagnostics when faced with protracted skin lesions, mainly if unresponsive to antibiotic therapy. Clinicians should be aware that *C. neoformans* can be responsible for unexplained skin lesions, even in healthy people. We recommend an oral antifungal therapy, for a case-dependent period, to achieve complete clinical and laboratory recovery of PCC.

## Figures and Tables

**Figure 1 diagnostics-13-03149-f001:**
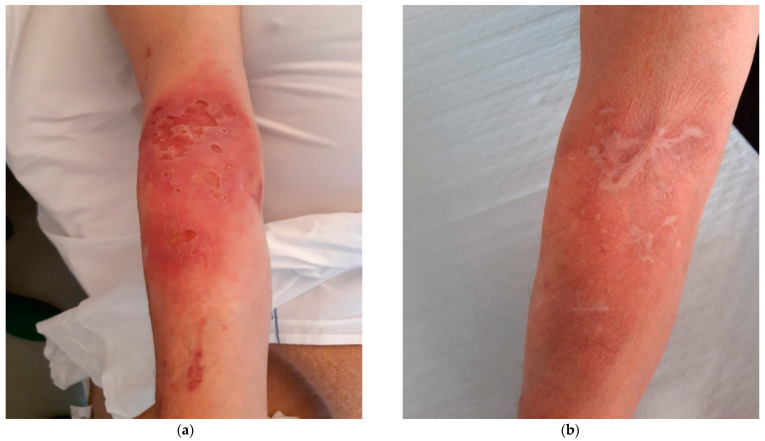
Evolution of skin lesions in a case of primary cutaneous criptococcosis (PCC): (**a**) the initial cutaneal lesion; (**b**) the outcome after 3 months of antifungal therapy and follow-up.

**Figure 2 diagnostics-13-03149-f002:**
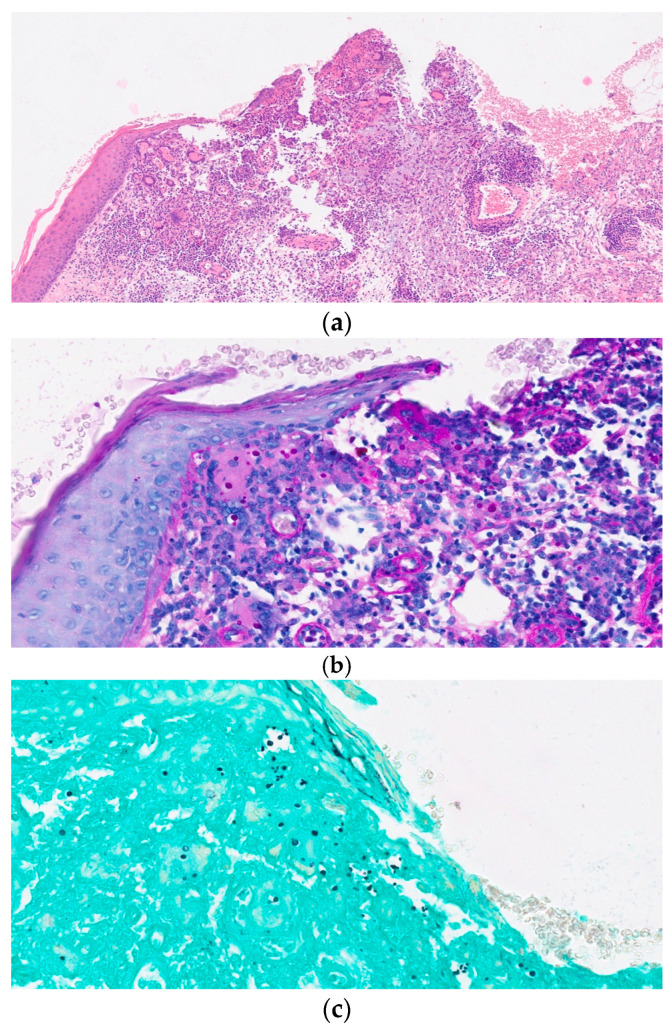
Histopathological findings of skin biopsy: (**a**) ulcerative, granulomatous inflammation within the dermis and subcutis—hematoxylin and eosin, original magnification (OM): 10×; (**b**,**c**) encapsulated, variably sized yeasts with thin walls are well demonstrated by special stain (**b**) periodic acid–Schiff stain, OM: 40× (**c**) Grocott–Gomori’s methenamine silver stain, OM. The histology picture showed a deep mycosis, characterized by a granulomatous suppurative and ulcerative inflammation in the dermis and subcutis with variably sized (3.5–8 μm in diameter) round to oval encapsulated yeasts in macrophages and giant cells. The cell wall of *Cryptococcus neoformans* was identified with gomori methenamine silver stain and PAS stain (Figure 2a–c).

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
