# Peer review of "Primary Cutaneous Cryptococcosis in an Immunocompetent Patient: Diagnostic Workflow and Choice of Treatment"

_diagnostics, 2023, doi:10.3390/diagnostics13193149_

Round 1

Reviewer 1 Report

The authors describe a 72-year-old man with a primary cutaneous cryptococcosis probably acquired in a small skin wound during outdoor work in a rural area.

The case is well described.

The English requires re-editing, both for grammar as well as orthography.

Reviewer 2 Report

Dear authors,

My general comments are positive:

Your paper is very nice written article and concerns important case with presentation of clinical features, diagnostics, treatment and monitoring of primary cutaneous cryptococcosis that was diagnosed according to newest classification.

It is desirable that such a case be published, to indicate the need for consideration of yeast infection especially when antibacterial treatment does not improve condition in patients. Moreover, interpretation of results is very important since patients with cutaneous cryptococcosis should be closely monitored since they are at increased risk of developing disseminated form or CNS cryptococcosis. On the contrary, secondary cryptococcal skin infection could implies dissemination of Cryptococcus spp.

I recommend it for publication with minor revision.

General comments

The part of paper dedicated to treatment of primary cutaneous cryptococcosis has to be write more clearly.

In the paper it is important to point out that in medical mycology diagnostic procedure is not difficult it is a problem because fungal infection is not considered.

Moreover, it will be useful to highlight that primary cutaneous cryptococcosis can be the source for occurrence of systemic cryptococcosis

P.S. Please, provide the official email of corresponding author

Moderate editing of English language required
